# The Genome of *Bacillus velezensis* SC60 Provides Evidence for Its Plant Probiotic Effects

**DOI:** 10.3390/microorganisms10040767

**Published:** 2022-04-01

**Authors:** Xiaoyan Dong, Chen Tu, Zhihong Xie, Yongming Luo, Lei Zhang, Zhaoyi Li

**Affiliations:** 1CAS Key Laboratory of Coastal Environmental Processes and Ecological Remediation, Yantai Institute of Coastal Zone Research, Chinese Academy of Sciences, Yantai 264003, China; xydong@yic.ac.cn (X.D.); ctu@yic.ac.cn (C.T.); ymluo@yic.ac.cn (Y.L.); leizhang@iae.ac.cn (L.Z.); lizhaoyichloe@163.com (Z.L.); 2National Engineering Research Center for Efficient Utilization of Soil and Fertilizer Resources, College of Resources and Environment of Shandong Agricultural University, Taian 271000, China; 3CAS Key Laboratory of Soil Environment and Pollution Remediation, Institute of Soil Science, Chinese Academy of Sciences, Nanjing 210008, China; 4Institute of Applied Ecology, Chinese Academy of Sciences, Shenyang 110016, China

**Keywords:** PGPR, root colonization, genome sequencing, *Bacillus*, soil-borne pathogens

## Abstract

Root colonization and plant probiotic function are important traits of plant growth-promoting rhizobacteria (PGPR). *Bacillus velezensis* SC60, a plant endophytic strain screened from *Sesbania cannabina,* has a strong colonization ability on various plant roots, which indicates that SC60 has a preferable adaptability to plants. However, the probiotic function of the strain SC60 is not well-understood. Promoting plant growth and suppressing soil-borne pathogens are key to the plant probiotic functions. In this study, the genetic mechanism of plant growth-promoting and antibacterial activity of the strain SC60 was analyzed by biological and bioinformatics methods. The complete genome size of strain SC60 was 3,962,671 bp, with 4079 predicted genes and an average GC content of 46.46%. SC60 was designated as *Bacillus velezensis* according to the comparative analysis, including average nucleotide polymorphism (ANI), digital DNA–DNA hybridization (dDDH), and phylogenetic analysis. Genomic secondary metabolite analyses indicated two clusters encoding potential new antimicrobials. The antagonism experiments revealed that strain SC60 had the ability to inhibit the growth of a variety of plant pathogens and its closely related strains of *Bacillus* spp., which was crucial to the rhizospheric competitiveness and growth-promoting effect of the strain. The present results further suggest that *B. velezensis* SC60 could be used as a PGPR strain to develop new biocontrol agents or microbial fertilizers.

## 1. Introduction

Due to the abuse of chemical fertilizers and pesticides in agricultural production, the quality of soil and agricultural products has been reduced, which also negatively affects the ecology and sustainability of agricultural development. Plant growth-promoting rhizobacteria (PGPR) are a kind of beneficial microbes that stably survive and colonize in the rhizosphere of plants. PGPR are the major source of biofertilizer strains, which show beneficial effects on crops, such as growth promotion, inhibition of soil-borne pathogens, and enhancement of plant tolerance [1]. The beneficial functions of PGPR on plants largely rely on root colonization ability, for which chemotactic motility and biofilm formation on the rhizoplane are the most important colonization processes [2,3]. Rhizospheric soil is rich in microbial communities, including microbes that exert plant growth-promoting properties [4]. Among the rhizosphere microbial communities, the *Bacillus* genus is among the most ubiquitous soil microbes and possesses several plant growth-promoting traits [5,6]. Members of the *Bacillus* genus are ubiquitous in soil due to their endospore-forming properties that allow them to survive in different ecological niches [7]. Members of the *Bacillus* genus use several mechanisms to improve plant growth and enhance plant pathogen resistance under various environmental conditions.

A key application of PGPR strains is to promote plant growth. Numerous species of *Bacillus* act as plant growth promoters, relying on multiple mechanisms that involve the synthesis of some phytohormones (auxin, cytokinin, and gibberellins), volatile compounds (VOCs), induction of 1-amino cyclopropane-1-carbocylate (ACC) deaminase, and fixation of atmospheric nitrogen [6,8,9,10]. For example, *Bacillus velezensis* FZB42 promoted *Triticum aestivum* root growth by secreting auxin [11]. In addition, *Bacillus* spp. also secrete phytase or acidic substances that help dissolve minerals (e.g., phosphorus and potassium) in the soil and thus enhance nutrient uptake by plants [12,13,14]. *Bacillus* spp. can also secrete siderophores to promote the absorption of iron by plants [15,16].

Another important application of PGPR is to protect the plant from pathogens by producing antimicrobial compounds or competing nutrients. *Bacillus velezensis*, a representative strain of the genus *Bacillus*, uses more than 8% of the genome to synthesize secondary metabolites in response to competing microbes in the plant rhizosphere [17]. A dozen non-ribosomal polypeptides (bacillaene, bacillibactin, macrolactin, bacillomycin D, fengycin, bacilysin, difficidin, and surfactin) with antifungal and antibacterial activities are synthesized by *Bacillus* spp. [18]. Biosynthetic gene clusters (BGCs) are responsible for the synthesis of various secondary metabolites [19]. Due to genetic variety, different strains of *Bacillus* exhibit unique differences in metabolites and antimicrobial capacity, and even among the closely related *Bacillus* spp. strains, there are many variations in the BGCs [20]. Therefore, based on the genome sequencing, it is possible to discover new BGCs encoding potential secondary metabolites by comparative analysis. This will provide a basis for screening new PGPR strains. Furthermore, these bacteria also can induce systemic resistance responses of plants to pathogens, improve abiotic stress tolerance, and suppress the growth of fungal and bacterial pathogens [21,22,23].

*Sesbania cannabina* is a multipurpose leguminous crop native to tropical and subtropical regions such as Asia, Africa, and Australia. Due to its wide adaptability and outstanding stress resistance to drought, waterlogging, and salinity, *Sesbania cannabina* is widely planted as a green manure to improve soil fertility [24,25]. In China, *Sesbania cannabina* as a pioneer plant has been successfully introduced to the saline-alkaline land restoration of the Yellow River Delta (YRD) [26]. Furthermore, *Sesbania cannabina* has strong disease and insect resistance, so it is an excellent material for isolating PGPR.

Strain SC60, which was isolated from the seeds of *Sesbania cannabina*, showed a strong colonization ability on different plant roots. However, the plant probiotic function of strain SC60 is unclear. This study intends to reveal the probiotic potential of the strain SC60 through the genome sequencing information mining, and to elucidate whether the strain SC60 can be used as a PGPR strain in agriculture through plant growth-promoting and pathogen-inhibiting experiments. Here, a series of gene clusters associated with promoting plant growth and inhibiting soil-borne pathogens were identified based on genome sequencing, and we confirmed that SC60 can promote plant root development and inhibit the growth of some pathogens. In addition, we further revealed that the strain SC60 can enhance its rhizosphere competitiveness by inhibiting the growth of closely related species. Thus, *B. velezensis* SC60 could be used as a PGPR strain to develop new microbial fertilizers.

## 2. Materials and Methods

### 2.1. Microbial Strains and Related Medium

The strains used in this study are described in Appendix A. *Bacillus* spp. were cultured in liquid LB medium (10 g tryptone, 5 g yeast extract, and 10 g NaCl per liter) with incubation at 37 °C for 24 h in a rotatory shaker (170 rpm). For secondary metabolite production, strain SC60 was grown in Landy medium (MgSO_4_·7H_2_O 0.5 g, KH_2_PO_4_ 1 g, KCl 0.2 g, MnSO_4_·H_2_O 10 mg, FeSO_4_·7H_2_O 5 mg, CuSO_4_·5H_2_O 0.2 mg, glucose 20 g, yeast extract 1 g, L-alanine 2 mg, L-glutamic 5 g per liter) at 30 °C and 170 rpm for 48 h. The bacterial pathogen *Ralstonia solanacearum,* preserved in the laboratory, was incubated at 30 °C in NA medium (10 g tryptone, 5 g yeast extract, and 5 g NaCl per liter). Fungal pathogens (*Fusarium*
*verticillioides*, *Phytophthora capsica*, *Fusarium oxysporum* sp. cucumebrium Owen, *Fusarium graminearum* schw, *Botrytis cinerea* Pers. ex Fr, *Aspergillus niger* Van Tiegh, *Fusarium solani*) preserved in the laboratory were incubated at 28 °C in PDA medium (200 g potato, 20 g glucose, and 20 g agar per liter).

### 2.2. Genomic DNA Preparation, Sequencing and Analysis

Genomic DNA was extracted from strain SC60 using the Bacteria Genomic DNA Extraction Kit (AxyGen, China, Hangzhou), and the genomic DNA of SC60 was sequenced at Shanghai Majorbio Biopharm Technology Co., Ltd. using an Illumina HiSeq×10 system and PacBio high-throughput sequencing technology. Quality control of the raw reads was performed, including base quality, base error rate, and base distribution. Raw Q20 is an important quality filtering parameter, and a total of 6,815,634 filtered paired-end reads with a length of 150 bp were obtained. The filtered reads were assembled using SOAPdenovo2 (http://soap.genomics.org.cn/, accessed on 20 September 2020) with a K-mer size of 17 bp to generate contigs [27]. The complete genome was assembled by Unicycler (v0.4.8) [28]. The coding sequence (CDS) of strain SC60 was predicted using Glimmer (http://ccb.jhu.edu/software/glimmer/index.shtml, accessed on 22 September 2020) [29]. Genome annotation of the CDS was accomplished using NR, Swiss-Prot, Pfam, COG, GO, and KEGG databases [30,31,32,33]. Gene islands (GIs) were predicted in the strain SC60 genome using IslandViewer (v1.2) [34].

### 2.3. Comparative Analysis of SC60 with Other Bacillus spp.

Whole genome sequences of Bacillus spp. were downloaded from the GenBank genome database (https://www.ncbi.nlm.nih.gov/genome/, accessed on 21 July 2021). ANI values were found using Jspecies WS (http://jspecies.ribohost.com/jspeciesws/, accessed on 29 July 2021). dDDH analysis was carried out by the Genome-to-Genome Distance Calculator (GGDC v3.0) (https://ggdc.dsmz.de/, accessed on 30 July 2021) [20]. Biosynthetic gene clusters for secondary metabolites were analyzed using antiSMASH (v4.0.2) [18]. Comparative genomic analysis was performed by BLAST Ring Image Generator software (BRIG) (https://sourceforge.net/projects/brig/, accessed on 21 July 2021) [35]. The phylogenomic neighbor-joining (NJ) tree was constructed based on the 2968 homologous genomes of 17 *Bacillus* spp., including *B.*
*subtilis*, *B. amyloliquefaciens*, and *B. velezensis* strains, with *B.*
*subtilis* as an outgroup. OrthoFinder_2.5.2 was used for homologous genome analysis, and MEGA software (v7.0.26) was used to construct a phylogenetic tree using the NJ method [36].

### 2.4. Root Colonization of Strain SC60

The colonization abilities of strain SC60 on sesbania, maize, tomato, and pepper root were conducted according to previously reported methods [37]. Briefly, we germinated the sterilized seeds in a sterile environment. The normally germinated seeds were then transferred into sterile vermiculite until two cotyledons emerged. The plants were washed with sterile water to remove the vermiculite on the surface of plant roots and transfer to 1/5 sterile Hoagland nutrient solution for culture. Overnight cultures of strain SC60 were transferred into 100 mL of LB medium at 1% inoculum and incubated at 37 °C and 170 rpm until an OD_600_ value of 1.0 was achieved. Cells were collected at 8000 rpm at 4 °C. The bacterial cells were washed in 1/5 Hoagland solution 3 times and then resuspended in sterile 1/5 Hoagland nutrient solution. Subsequently, the strains were transferred to sterile seedlings with 3 true leaves to a final concentration of 0.5 × 10^6^ cells mL^−1^. After three days of cultivation, the roots of plants were washed with sterilized water, and the cells colonized on roots were collected by shaking and measured by plate counting.

### 2.5. Plant Growth-Promoting Assay of Strain SC60

*Sesbania cannabina* seeds were surface-sterilized and germinated in sterile conditions. The normally germinated seeds were then transferred to a square plate covered with sterile filter paper. Then, 15 mL of 1/5 Hoagland solution containing a final concentration of 0.5 × 10^6^ cells mL^−1^ was added to the treatment group, while 15 mL of sterile 1/5 Hoagland solution was added to the control group. The plates were incubated in a climatic chamber at 12 h light (26 °C)/12 h dark (22 °C) and 70% relative humidity. After seven days of cultivation, root development was recorded.

### 2.6. Plant Pathogens’ Growth Inhibition Assays

The antagonistic activities of strain SC60 against fungal and bacterial pathogens were measured in vitro on PDA and NA agar, respectively. Fungi preserved at −80 °C were activated on PDA plates; after spore germination, the fungal pieces were transferred to new culture medium with an inoculation shovel to completely restore their activity. The antagonistic fungal activity was carried out on 9 cm plates. First, the strain SC60 was activated in LB medium until the OD_600_ reached 1.0, and then a 5 mm-diameter fungal piece was inoculated onto the center of a PDA plate. Then, 2.5 µL of strain SC60 suspension was pipetted onto 2 cm from each of the fungal pieces, and the plates were incubated for 5 days at 28 °C. The antagonistic effect of strain SC60 against fungal pathogens was evaluated by whether there was an inhibition zone [38]. The activity of antagonistic bacteria was detected by a similar method. All antagonistic experiments were performed in triplicate.

### 2.7. Closely Related Species’ Growth Inhibition Assays

The strain SC60 was fermented in Landy medium and the culture was collected by centrifugation at 8000 rpm for 10 min. The supernatant was dehydrated in a freeze dryer, and then extracted five times with methanol (100 mL of methanol to extract 1 L of supernatant). The extraction was filtered by a 0.22 μm sterile filter. SC60 were detected on LB medium, and a sterile Oxford cup with an average of 150 μL extraction was placed on a plate with the tested *Bacillus* spp. strains, and 150 μL methanol as CK. The plates were stored at room temperature for 36 h [17].

## 3. Results and Analyses

### 3.1. Comparative Genome Analysis of Strain SC60 with Bacillus Strains

The genomic structure information showed that the chromosome length of strain SC60 was 3,962,671 bp, and a BLAST comparison with *B. velezensis* and *B. subtilis* strains is shown in Figure 1. The SC60 genome is composed of a single circular chromosome with a length of 3,962,671 bp, and the average G + C content is 46.46%. The whole genome of SC60 was predicted to contain 4079 coding sequences (CDSs), 27 rRNA, 89 tRNA, and 83 sRNA genes (Table 1). The classification of SC60 genes into clusters of orthologous groups (COGs) assigned 2986 CDSs to at least one COG group (73.2%). A total of 1093 CDSs were not classified into COGs. PHAST analysis of the strain SC60 genome identified 3 prophage regions (Table 1) depicted on the green ring (Figure 1), which represents 3.6% of the entire genome. Sixteen gene islands (GIs) are depicted on the blue ring in Figure 1, which represent 3.3% of the entire genome.

In general, it is inaccurate to distinguish *B. subtilis, B. amyloliquefaciens*, and *B. velezensis* according to 16 S rRNA gene sequences. At present, distinguishing and detecting novel species in *Bacillus* taxa is mostly based on phylogenetic analysis of the core genome. To identify the species of strain SC60 in *Bacillus*, the phylogenomic NJ tree was constructed based on 2968 homologous genomes. The comparative phylogenomic analysis indicated that strain SC60 showed the highest homology with *Bacillus velezensis* (Figure 2). Indeed, ANI and dDDH analyses between strain SC60 and *Bacillus velezensis* were ≥97.45% and ≥91.20%, respectively, which indicated that the strain SC60 was finally identified as *Bacillus velezensis* (Table 2).

### 3.2. Plant Growth-Promoting Traits of SC60

*Bacillus* spp. is a Gram-positive soil bacterium that is commonly found in the rhizosphere environment, the area of soil surrounding plant roots. Root colonization of microbes determines the efficiency of their beneficial effects on plants. In this study, the colonization ability of the strain on the root of *Sesbania cannabina*, maize, pepper, and tomato was tested, and the results indicated that strain SC60 can colonize the root surface of many plants (Figure 3A). Moreover, we found that strain SC60 can promote the root development of *Sesbania cannabina* (Figure 3B,C). Further genomic analysis showed that strain SC60 contained multiple genes related to root colonization and plant growth promotion, and it has even been postulated that strain SC60 has potential as a biocontrol bacterium in the rhizosphere (Table 3).

### 3.3. Secondary Metabolite Cluster Analysis

Thirteen BGCs of secondary metabolites were identified in the SC60 genome by the antiSMASH tool (Table 4), including two encoding NRPS, two transAT-PKS, three transATPKS-NRPS, one ladderane, two terpene, one lantipeptide, one T3PKS, and one PKS-like. This analysis showed that about 19% of the SC60 genome is responsible for the regulation, biosynthesis, and transport of antimicrobials. Seven clusters were clearly identified to participate in the synthesis of surfactin, plantazolicin, macrolactin, bacillaene, difficidin, bacillibactin, and bacilysin. Furthermore, Cluster 7 is responsible for the synthesis of both bacillomycin D and fengycin. All clusters have antimicrobial/antibacterial activities or are related to antibiosis. For the 8 clusters, compared with the reference strain *B. velezensis* FZB42, it was found that the gene cluster arrangement of SC60 was very similar to that of FZB42 (Figure 4, Table 4). The secondary metabolite cluster in strain SC60 was compared with the cluster of four *Bacillus* strains (FZB42, SQR9, DSM7, and 168) (Table 4), and we found cluster 3 and cluster 12 encoding potential novel metabolites. Cluster 3, encoding PKS-like, showed 7% gene similarity with butirosin A, but the biosynthetic genes of strain SC60 are inconsistent with the butirosin A biosynthetic genes in the MIBiG database. Cluster 12, encoding lantipeptide class II, was not present in any of the four *Bacillus* strains.

### 3.4. Antagonistic Activity against Plant Pathogens

To explore the control potential of strain SC60 against soil-borne plant pathogens, the antagonistic activity was determined in PDA and NA media. The strain SC60 was cultured with *Fusarium*
*verticillioides*, *Phytophthora capsica*, *Fusarium oxysporum* sp. cucumebrium Owen, *Fusarium graminearum* schw, *Botrytis cinerea* Pers. ex Fr, *Aspergillus niger* Van Tiegh, and *Fusarium solani*, and the results showed that the strain SC60 could significantly inhibit the growth of these fungal pathogens. Moreover, strain SC60 also strongly inhibited the growth of the plant pathogenic bacterium *Ralstonia solanacearum*. This result indicated that strain SC60 exhibited a broad spectrum against soil-borne plant pathogens (Figure 5).

### 3.5. Antagonistic Activity against Bacillus spp.

Plant roots can secrete a variety of root exudates to provide nutrition for the growth of microbes and allow beneficial microbes to grow and colonize. However, the nutrition provided by plant root exudates is limited. Beneficial bacteria should not only compete with pathogenic bacteria but also compete with closely related strains because of the similar nutritional utilization among the closely related strains. To enhance the rhizosphere competitiveness of PGPRs, some strains have evolved to produce a variety of antibiotics that inhibit or kill other rhizosphere microbes. In this study, the antagonistic ability of strain SC60 against closely related species was verified by the Oxford cup method. The results suggested that the fermentation broth of strain SC60 significantly inhibited the growth of a variety of *Bacillus* spp., including *Bacillus megaterium*, *Bacillus amyloliquefaciens*, *Bacillus*
*velezensis*, *Bacillus cereus*, *Bacillus halotolerans*, *Bacillus*
*brevis, Bacillus fordii*, and *Bacillus subtilis* (Figure 6). This characteristic can significantly enhance the rhizosphere competitiveness of strain SC60 and play a major role in rhizosphere biocontrol.

## 4. Discussion

PGPR are the major source of biofertilizer strains, which show beneficial effects on crops, such as growth promotion and inhibition of soil-borne pathogens. However, their beneficial functions largely depend on efficient colonization on roots, for which chemotactic motility and biofilm formation on the rhizoplane are the most important colonization processes. There is a *fla-che* operon responsible for chemotactic motility and seven methyl-accepting chemotaxis proteins in the SC60 genome. Biofilm matrices are controlled by several synthesis genes, such as EPS, which is synthesized by the *epsA-O* operon [39]. The amyloid fibers, encoded by the *tapA*-*sipW*-*tasA* operon, are primarily composed of the TasA protein [40]. BslA is a biofilm-surface layer protein, which is responsible for the hydrophobic layer on the biofilms’ surface encoded by *bslA* [41]. In addition, γ-polyglutamate plays an important role in enhancing the structure of biofilms [42]. The genome of strain SC60 contains the genes encoding the extracellular matrix of the biofilm. The beneficial effect of rhizospheric *Bacillus* results from the synergy of multiple factors, including the production of phytohormones and the secretion of phytase and the volatile compound acetoin [43,44,45]. The genome of SC60 contains the coding gene cluster *trpABFCDE* of tryptophan, the precursor of IAA synthesis, the coding gene *phy* of phytase, and the coding gene cluster *alsDSR* of acetoin. BGCs are responsible for the synthesis of various secondary metabolites, which contribute to the competitive role between different microbes and resistance to self-encoded antibiotics [46]. In the soil environment, there are various pathogenic fungi and bacteria as well as a large number of closely related species of biocontrol strains. However, due to the lack of nutrition around the plant rhizosphere, PGPR strains compete with these strains. Therefore, PGPR strains should have strong rhizosphere competitiveness. *Bacillus* spp. devote a large percentage of their genomes to encoding bioactive secondary metabolites that are crucial to competition in the rhizosphere. A large number of secondary metabolites with antibacterial activity were reported in *Bacillus*
*velezensis* strain FZB42 to inhibit plant pathogens. For example, surfactin, fengycin, and bacillomycin D have strong growth-inhibitory activities against cucumber *Fusarium* wilt [47]. The known secondary metabolite gene cluster in SC60 was observed to be similar to the corresponding gene cluster in FZB42, with high homology at the nucleotide level (Figure 4). This shows that strain SC60 also has the potential to prevent and control a variety of plant diseases.

Competition with closely related species also plays a crucial role in the effective biocontrol function of PGPR strains. In recent years, it has been found that the plantazolicin and amylocyclicin produced by *B. velezensis* FZB42 inhibited the growth of *Bacillus megaterium* and *Bacillus subtilis*. *Bacillus velezensis* SQR9 was also found to encode novel antimicrobial fatty acids to inhibit the growth of closely related *Bacillus* [17]. Recent studies have shown that the BGCs distance of the *Bacillus* genome is significantly positively correlated with their phylogenetic distance; moreover, the interspecies antagonism of *Bacillus* spp. was also significantly positively correlated with the distance of BGCs, suggesting that BGCs profiling may play a role in regulating interspecies antagonism [48]. Therefore, we can speculate that the antagonism between the same *Bacillus* species should be weak. In the present study, the strain SC60 encoded 13 BGCs, indicating that it is rich in secondary metabolites. Additionally, we found that strain SC60 can not only inhibit the growth of distantly related strains such as *Bacillus megaterium* and *Bacillus cereus*, but also inhibit the growth of closely related strains such as *Bacillus amyloliquefaciens* and *Bacillus*
*velezensis* (FZB42 and SQR9) (Appendix A). The results imply that strain SC60 may produce a new antibacterial compound. In the future, we will further excavate the coding gene cluster in strain SC60 that inhibits the growth of closely related strains, and determine the properties and structure of the secondary metabolite through chemical methods. Although the secondary metabolites of strain SC60 can inhibit the growth of many kinds of *Bacillus*, according to the existing research, we speculated that some *Bacillus* species that are distantly related to strain SC60 may also have inhibition on the growth of strain SC60. Our findings will provide an important theoretical basis for the agricultural application of strain SC60.

## 5. Conclusions

In this study, the colonization ability of the *Bacillus velezensis* SC60 on the root surface of various plants was tested and the effects of SC60 on the root development of *Sesbania cannabina* were analyzed. We found that the *Bacillus velezensis* SC60 has strong colonization and growth-promoting functions. This research further revealed the probiotic mechanism of the *Bacillus velezensis* SC60 through genomic analysis and antagonism experiments. In addition, the results of the study also showed that the *Bacillus velezensis* SC60 can enhance its rhizosphere competitiveness by inhibiting the growth of closely related species. Our findings provide a potential resource for biocontrol agents or biofertilizers.

## Figures and Tables

**Figure 1 microorganisms-10-00767-f001:**
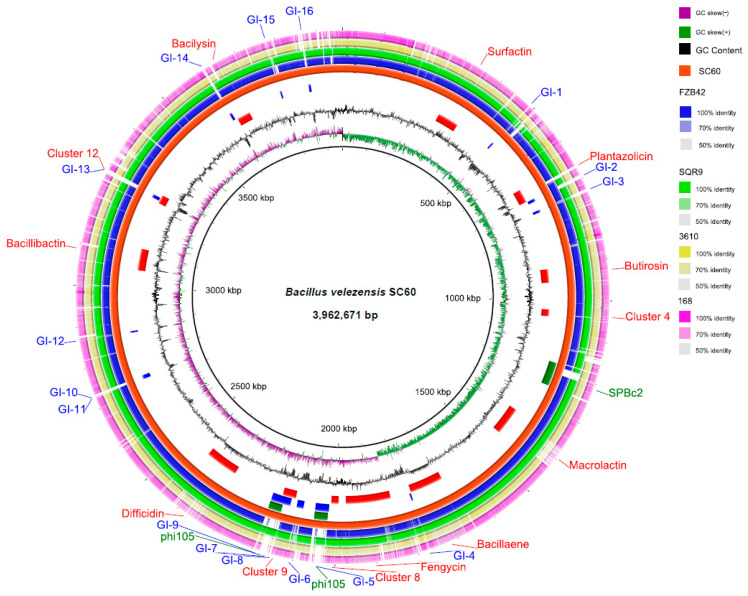
Whole genome features and comparative genome map of *Bacillus velezensis* SC60. Circle map displays from the inside to outside: (1) GC Skew, (2) GC Content, (3) secondary metabolite clusters, (4) gene islands, (5) predictive prophage clusters, (6) genome SC60, and (7, 8, 9, 10) blast comparison of the SC60 genome with *Bacillus* strains FZB42, SQR9, 3610, and 168, respectively. Map and comparative genomic analysis were performed with BRIG.

**Figure 2 microorganisms-10-00767-f002:**
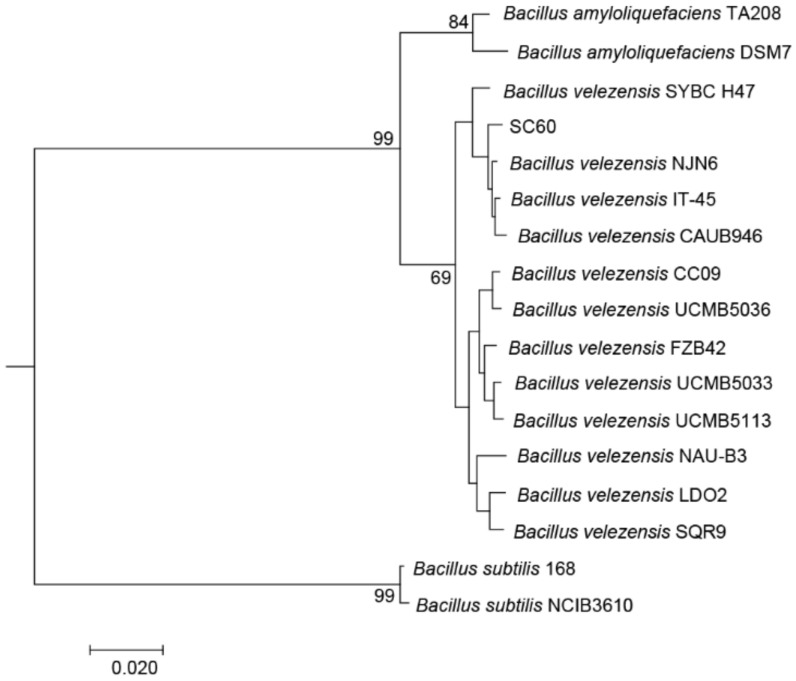
Phylogenetic tree based on the core genomes of 17 *B. subtilis*, *B. amyloliquefaciens*, and *B. velezensis* strains. OrthoFinder_2.5.2 was used for homologous genome analysis, and MEGA 7 software was used to construct a phylogenetic tree using the NJ method.

**Figure 3 microorganisms-10-00767-f003:**
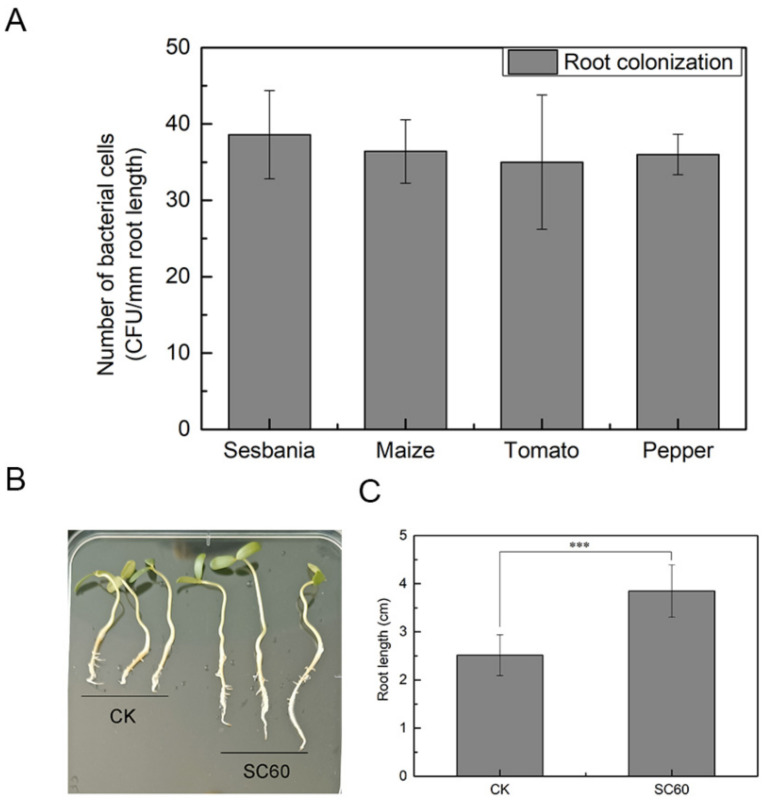
Colonization and root growth-promoting ability of strain SC60. (**A**) Root surface colonization of strain SC60. Error bars indicate the standard deviations based on five independently replicated experiments. (**B**) Effects of strain SC60 on root development of *Sesbania cannabina*. CK is the control group. The length of root is measured and the statistics are shown in Figure C (**C**) Quantitative data on growth of *Sesbania cannabina*. Error bars indicate the standard deviations based on six independently replicated experiments and asterisks (***) above the bars indicate significant differences (*p* < 0.001).

**Figure 4 microorganisms-10-00767-f004:**
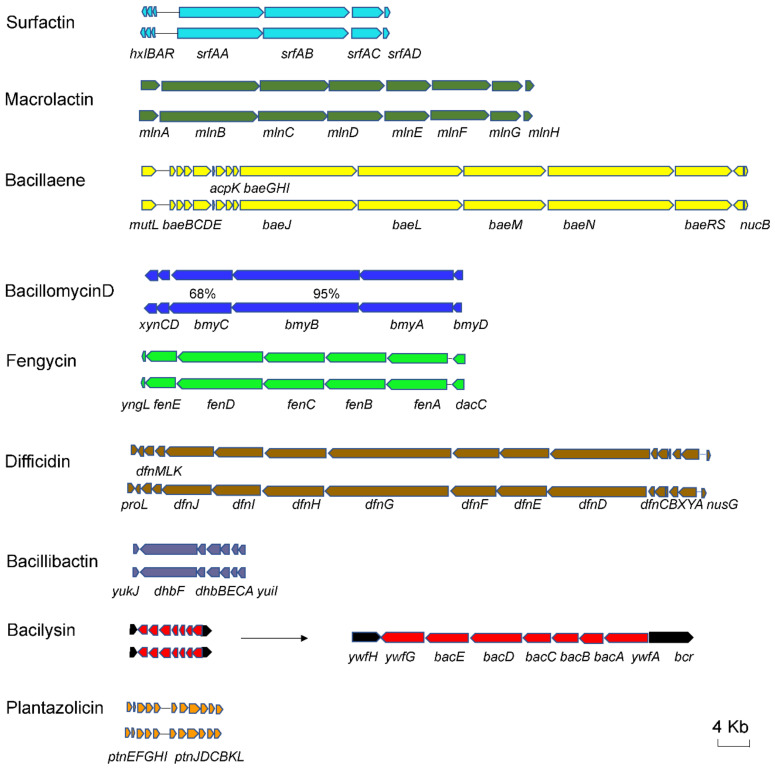
Comparisons of NRPS, TransATPKS, and lantipeptide clusters in SC60 (above) and FZB42 (below). The percentage indicates less than 98% identity between strain sequences.

**Figure 5 microorganisms-10-00767-f005:**
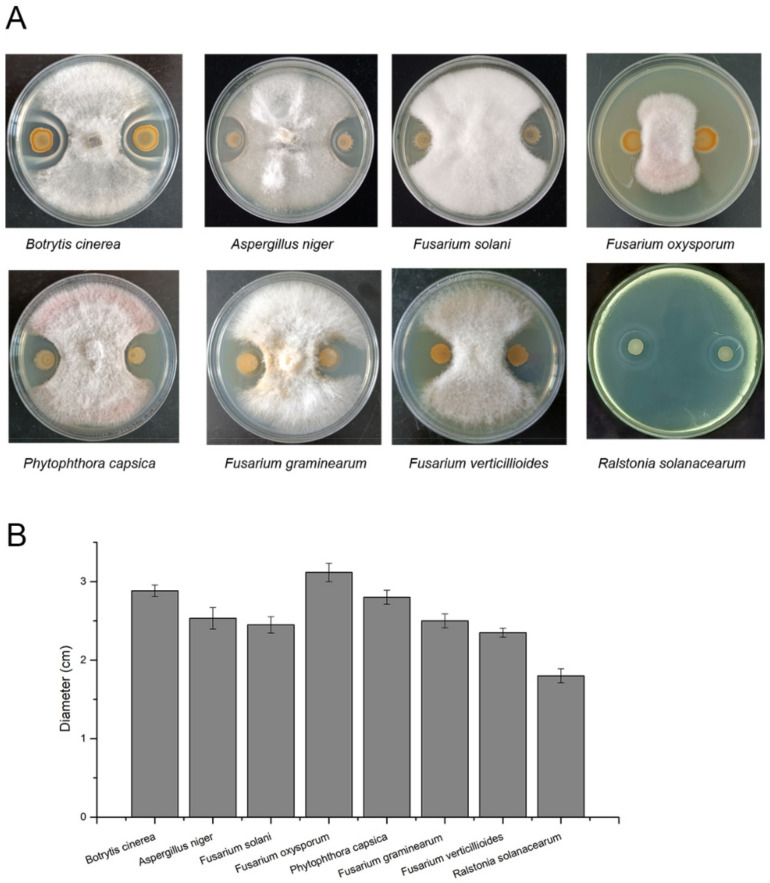
Antagonistic activity of strain SC60 against eight plant pathogens. (**A**) Inhibition effect of strain SC60 on the growth of plant pathogens. (**B**) The size of the zone of inhibition. Error bars indicate the standard deviations based on six independently replicated experiments.

**Figure 6 microorganisms-10-00767-f006:**
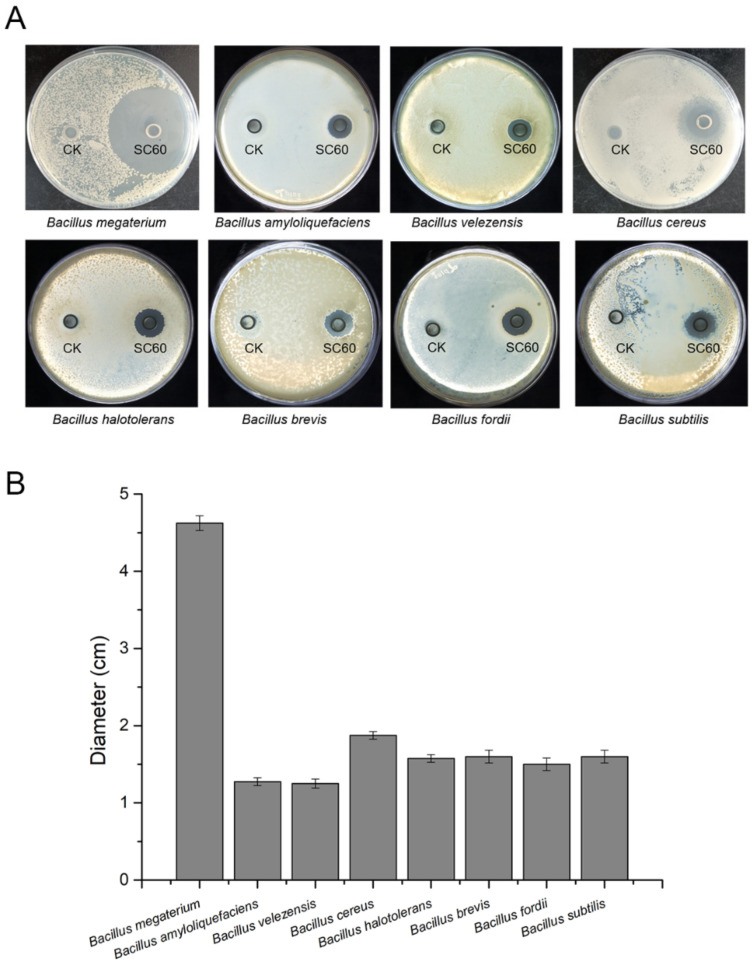
Antagonistic activity of strain SC60 against closely related *Bacillus* spp. (**A**) Inhibition effect of strain SC60 on the growth of *Bacillus* spp. (**B**) The diameter of the zone of inhibition. Error bars indicate the standard deviations based on four independently replicated experiments.

**Table 1 microorganisms-10-00767-t001:** General genomic features of *Bacillus velezensis* SC60.

Feature	*B. velezensis* SC60
Genome (bp)	3,962,671 bp
G + C (%)	46.46%
CDS number	4079
Average gene length (bp)	863.62
rRNA	27
tRNA	89
sRNA	83
Genes assigned to COG	2896
prophage regions	3

**Table 2 microorganisms-10-00767-t002:** Comparative genomic analysis of *Bacillus velezensis* SC60 with *Bacillus* genomes.

Strains	GenBank Accession No.	ANI (%)	dDDH (%)	GC (%)	Size (bp)
*Bacillus velezensis* SC60	NZ_CP072311	100	100	46.46	3,962,671
*Bacillus velezensis* NJN6	NZ_CP007165	99.21	96.80	46.60	4,052,546
*Bacillus velezensis* CAU B946	NC_016784.1	99.11	97.50	46.50	4,019,861
*Bacillus velezensis* IT-45	NC_020272	98.99	97.00	46.62	3,928,857
*Bacillus velezensis* SYBC_H47	NZ_CP017747	98.15	93.60	46.40	3,884,433
*Bacillus velezensis* UCMB5036	NC_020410.1	97.68	94.90	46.60	3,910,324
*Bacillus velezensis* UCMB5113	NC_022081	97.63	94.20	46.70	3,889,532
*Bacillus velezensis* LDO2	NZ_CP029034.1	97.60	94.30	46.50	3,947,271
*Bacillus velezensis* NAU-B3	NC_022530.1	97.52	91.20	45.99	4,196,170
*Bacillus velezensis* FZB42	NC_009725	97.52	93.80	46.50	3,918,596
*Bacillus velezensis* SQR9	NZ_CP006890	97.49	92.10	46.10	4,117,023
*Bacillus velezensis* CC09	NZ_CP015443	97.49	91.80	46.10	4,167,153
*Bacillus velezensis* UCMB5033	NC_022075	97.45	92.10	46.20	4,071,167
*Bacillus amyloliquefaciens* DSM_7	NC_014551	93.29	78.50	46.10	3,980,199
*Bacillus amyloliquefaciens* TA208	NC_017188	93.23	78.70	45.80	3,937,511
*Bacillus subtilis* 168	NZ_CP010052	76.26	28.70	43.50	4,215,619
*Bacillus subtilis* NCIB3610.	NZ_CP020102	76.26	28.70	43.50	4,215,607

**Table 3 microorganisms-10-00767-t003:** PGPR trait-associated genes identified in the SC60 genome.

PGPR Traits	Gene ID	Gene Name	Function
Colonization traits			
Chemotaxis and motility	gene0785	*yfmS*	methyl-accepting chemotaxis protein
	gene1113	*hemAT*	methyl-accepting chemotaxis protein
	gene1482	*mcpC*	methyl-accepting chemotaxis protein
	gene3157	*tlpB*	methyl-accepting chemotaxis protein
	gene3158	*mcpA*	methyl-accepting chemotaxis protein
	gene3159	*tlpA*	methyl-accepting chemotaxis protein
	gene3160	*mcpB*	methyl-accepting chemotaxis protein
	gene1720–1751	*fla-che* operon	Flagellar biosynthesis, chemotaxis protein
Biofilm	gene3492–3506	*epsA-O*	polysaccharide biosynthesis protein
	gene2571–2573	*tapA-sipW-tasA*	amyloid fibers biosynthesis protein
	gene3142	*yuaB*	biofilm-surface layer protein
	gene3657–3659	*pgsA-C*	γ-poly-glutamate biosynthesis protein
Quorum sensing	gene3211–3214	*comAPXQ*	Quorum sensing
	gene3093	*luxS*	Autoinducer 2 (AI-2) synthesis protein
plant-growth-promoting factors			
Indole-3-acetic acid (IAA)	gene2346–2351	*trpABFCDE*	tryptophan biosynthesis operon
	gebe3897	*ysne*	tryptophan acetyltransferase
phytase	gene2162	*phy*	phytase biosynthesis protein
acetoin	gene3672–3674	*alsDSR*	acetoin biosynthesis operon

**Table 4 microorganisms-10-00767-t004:** Comparative analysis of secondary metabolite clusters of *Bacillus velezensis* SC60 identified in the genome with reference genomes.

*Bacillus velezensis* SC60						Presence (+) or Absence (−) of Secondary Metabolites Clusters in *Bacillus* Strains			
Cluster Number	Synthetase	Predicted Large Cluster Position	bp	Metabolites	MIBiG ID (% of Genes Show Similarity)	FZB42	SQR9	DSM7	168
1	nrps	305,409–370,816	65,407	Surfactin	BGC0000433 (82%)	+	+	+	+
2	ladderane	647,464–688,663	41,199	Plantazolicin	BGC0000569 (91%)	+	−	−	−
3	PKS-like	905,286–946,530	41,244	Butirosin	BGC0000693 (7%)	+	+	+	−
4	terpene	1,028,575–1,049,315	20,740	-	-	+	+	+	−
5	transatpks	1,354,640–1,440,527	85,887	Macrolactin	BGC0000181 (100%)	+	+	−	−
6	transatpks-nrps	1,666,678–1,769,379	102,701	Bacillaene	BGC0001089 (100%)	+	+	+	+
7	transatpks-nrps	1,833,815–1,971,640	137,825	Fengycin	BGC0001095 (100%)	+	+	+	+
8	terpene	1,994,479–2,016,362	21,883	-	-	+	+	+	+
9	t3pks	2,126,023–2,167,121	41,098	-	-	+	+	+	+
10	transatpks	2,326,876–2,427,336	100,460	Difficidin	BGC0000176 (100%)	+	+	−	−
11	bacteriocin-nrps	3,053,922–3,120,713	66,791	Bacillibactin	BGC0000309 (100%)	+	+	+	+
12	lantipeptide	3,269,802–3,296,120	26,318	-	-	−	−	−	−
13	NRPS	3,626,355–3,667,771	41,416	Bacilysin	BGC0001184 (100%)	+	+	+	+

## Data Availability

The assembled genome sequences for strain SC60 were uploaded to the NCBI GenBank, project number PRJNA716758.

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
