# Peer review of "The Genome of Bacillus velezensis SC60 Provides Evidence for Its Plant Probiotic Effects"

_microorganisms, 2022, doi:10.3390/microorganisms10040767_

Round 1
Reviewer 1 Report
This manuscript describes the characterisation, genetically and microbiologically, of a novel strain of B. velezensis. This strain has potentially useful antimicrobials and economically useful traits as a plant growth promoting and protecting beneficial bacteria. The study is well done and covers the key important aspects for characterising this isolate. The English is clear and comprehensible. In some cases the English is not perfect, but in no cases does this create confusion to the reader. This manuscript is a useful addition to current knowledge of beneficial plant bacteria.
Reviewer 2 Report
Dong et al. studied the effects of B. velezensis SC60 on plants and soil pathogens. They assembled the complete genome of SC60 and presented the phylogenetic analysis as well as other characteristics of the genome. They found that SC60 promoted growth of S. cannabina, and inhibited plant pathogens and closely related strains. Generally I think the project was well done and have mostly minor comments. The biggest weakness for me in this paper is the lack of quantitative results - I would much prefer to see actual number results instead (or in addition to) photos.
Abstract:
Good
Intro:
Line 35: probably should define PGPR again here even if it’s already defined in abstract
Line 37: space needed before [1] and many other references throughout the paper, check and fix
Line 43: Members of the Bacillus genus…
Line 45: Members of the Bacillus genus use…
Line 52: secrete
Line 54: Bacillus spp.
Line 65: Can you provide a bit more background on Sesbania cannabina? you say it is multipurpose - I suggest writing what those purposes are as well as provide some more information about the economic and other importance of this crop. this will help readers understand the impact of your work
Lines 71-72: some text appears to be highlighted with a very faint gray, check and fix.
Methods:
Line 95: State exactly what quality filtering parameters you used
Line 98-101: Softwares and databases should be cited
Line 98: how were 2 programs used? did you check which was better and choose one of them?
Line 99-100: again, why were multiple programs used for CDS prediction? You could probably just use Prodigal…
Line 107: delete “the” before antiSMASH
Line 112: need to mention which plants the root colonization tests were performed on
Line 113: Briefly, we germinated…
Line 127: 15 mL 1/5 Hoagland solution containing a final concentration of 0.5 x 10^6 cells mL-1 was added to the treatment group while 15 mL sterile Hoagland solution was added to the control group.
Line 129: in a climatic chamber
Line 129: need to state the conditions of the climatic chamber, temperature, humidity, light cycle
Line 133: Fungi preserved at -80ËšC were activated on PDA plates; …
Line 138: delete “The”
Line 150: for 36 h
Results:
Line 160: change “in at least” to “to at least”
Line 161: COGs
Line 164: The gene island methods/program should be mentioned in the methods not results
Line 166: Figure 1. Is it possible to get any higher resolution for this figure? Why is there a red ring and SC60 in the legend? If the genome is SC60, then of course the whole thing is going to be 100%. This ring and legend entry could be removed. Velezensis should be lowercase in the middle of the circle.
Line 175: change “taxa” to “species”
Line 181: would like some more details on the tree methods here or in methods section. is it a concatenated alignment? how many genes used?
Line 187: Bacillus spp. are gram-positive soil bacteria that are commonly…
Line 190: This needs to be in the methods - you never stated that you tested all of these plants!
Line 193: delete “information”
Line 197: Figure 3B - did you collect any quantitative data on growth? This would be good to present.
Line 234: Figure 5 - again, did you collect any quantitative data? It would be good to measure and present the size of the zone of inhibition.
Line 250: SC60 not SC6. Again, it would be good to measure and present the size of the zone of inhibition. Clarify what CK is.
Discussion:
Line 272: Bacillus spp. devote a larger percentage of their genomes…
Line 281: what about inhibition of SC60? You show that SC60 inhibits others but surely others also inhibit SC60. Did you study this? Do you have any comment to add to the text about this?
Line 288: delete “substance”
Conclusions:
Good
Reviewer 3 Report
The manuscript presented for review is written in a simple and clear manner. I have included some minor remarks in pdf. I would change the title - it's too long. I do not fully agree that the entire genome was studied, but the focus was more on the metabolites produced and the difference between them. In the introduction, there is no clearly stated goal and research assumptions. How is genome-wide research going to answer any antagonistic features ?? Why not expressing single genes? Such information should also be included in the introduction. Yes, there is a meaning of Bacillus taxa, but are only these species so important? 2.3 - how was the tree correctness checked? How long was the maintenance used? Why is there no outgroup? Where are the sequences from - authors or databases coming from, why is there no comparison with other sequences deposited by other authors? I like the photos presented. The results are presented clearly. Very poor discussion that could be linked to the results.

Round 2
Reviewer 3 Report
The presented version of the manuscript has been corrected taking into account my suggestions. In line 155 there is no preserved space called Fungi (yellow). I have no further comments.